# Medical Interventions for Patellofemoral Pain and Patellofemoral Osteoarthritis: A Systematic Review

**DOI:** 10.3390/jcm9113397

**Published:** 2020-10-23

**Authors:** Erin M. Macri, Harvi F. Hart, David Thwaites, Christian J. Barton, Kay M. Crossley, Sita M.A. Bierma-Zeinstra, Marienke van Middelkoop

**Affiliations:** 1Department of General Practice, Erasmus University Medical Center, Doctor Molewaterplein 40, 3015 GD Rotterdam, The Netherlands; s.bierma-zeinstra@erasmusmc.nl (S.M.A.B.-Z.); m.vanmiddelkoop@erasmusmc.nl (M.v.M.); 2Department of Family Practice, The University of British Columbia, 5950 University Boulevard, Vancouver, BC V6T 1Z3, Canada; 3Sport and Exercise Medicine Research Centre, La Trobe University, Health Sciences Building 3, Bundoora VIC 3086, Australia; h.hart@latrobe.edu.au (H.F.H.); david@completesportscare.com.au (D.T.); christian@completesportscare.com.au (C.J.B.); K.Crossley@latrobe.edu.au (K.M.C.); 4Department of Physical Therapy, Western University, London CA N6G 2V4, UK; 5Department of Surgery, St Vincent’s Hospital, University of Melbourne, Building 181, Grattan St, Melbourne VIC 3010, Australia

**Keywords:** patellofemoral pain, patellofemoral osteoarthritis, pharmaceuticals, nutraceuticals, surgery

## Abstract

Patellofemoral pain (PFP) and patellofemoral osteoarthritis (PFOA) are common, persistent conditions that may lie along a pathological spectrum. While evidence supports exercise-therapy as a core treatment for PFP and PFOA, primary care physicians commonly prescribe medication, or refer for surgical consults in persistent cases. We conducted a systematic review of medical interventions (pharmaceutical, nutraceutical, and surgical) for PFP and PFOA to inform primary care decision making. Methods: Following protocol registration, we searched seven databases for randomized clinical trials of our target interventions for PFP and PFOA. Our primary outcome was pain. We assessed risk of bias, calculated standardized mean differences (SMDs) and determined the level of evidence for each intervention. Results: We included 14 publications investigating pharmaceutical or nutraceutical interventions, and eight publications investigating surgical interventions. Two randomized control trials (RCTs) provided moderate evidence of patellofemoral arthroplasty having similar pain outcomes compared to total knee arthroplasty in isolated PFOA, with SMDs ranging from −0.3 (95% CI −0.8, 0.2, Western Ontario McMaster Pain Subscale, 1 year post-surgery) to 0.3 (−0.1, 0.7, SF-36 Bodily Pain, 2 years post-surgery). Remaining studies provided, at most, limited evidence. No efficacy was demonstrated for oral nonsteroidal anti-inflammatories or arthroscopic surgery. Conclusions: Pharmaceutical and nutraceutical prescriptions, and surgical referrals are currently being made with little supporting evidence, with some interventions showing limited efficacy. This should be considered within the broader context of evidence supporting exercise-therapy as a core treatment for PFP and PFOA.

## 1. Introduction

Patellofemoral pain (PFP) is a condition characterized by diffuse peri- or retro-patellar pain that is made worse by activities that increase patellofemoral joint load such as squatting, negotiating stairs, or running [1]. PFP is associated with reduced physical activity [2,3], impaired quality of life [4,5], and psychosocial features such as anxiety and depression [3,6]. PFP may also present with structural findings on imaging; particularly in older adults with patellofemoral osteoarthritis (PFOA) [7,8,9]. Annual prevalence of PFP in the general population is estimated at almost 23% [10,11], and it is the most common knee complaint seen by general practitioners [12]. At least 1.5% of over 30 million insured individuals who seek outpatient orthopedic care in the United States do so for PFP [13].

PFP is frequently persistent. In prospective studies of individuals diagnosed with PFP, 40–77% report persistent pain 6 to 20 years later [14,15]. There is an emerging hypothesis that PFOA may represent a long-term sequela of persistent PFP [7,8,16,17,18]. Like PFP, PFOA is associated with pain [9,19], reduced physical function [19], and lower quality of life [20], and PFOA in turn may lead to whole knee osteoarthritis (OA) [21]. These long-term consequences implore clinicians and researchers to ensure that the best evidence informs patient management, and highlights an urgent need for research aimed at improving patient outcomes.

Primary care clinicians prescribe pharmaceuticals to 25% or more of patients presenting with knee complaints such as PFP and PFOA, with underlying inflammation or structural features presumed to cause symptoms [12,22]. These prescriptions may be made in combination with exercise advice or referrals to physical therapy [12], which are recommended for evidence-based management of PFP and PFOA [23,24,25,26,27]. In addition to pharmaceutical agents, nutraceuticals are reported to be of therapeutic benefit in certain musculoskeletal conditions [28]. They are derived from dietary sources, can be administered in different formats (e.g., taken orally or by injection), and are regulated differently in different countries. Therefore, nutraceuticals can be prescribed and administered in clinical settings, but in many countries patients can also obtain nutraceuticals off-the-shelf and taken without specific recommendations by a medical practitioner. Finally, in cases where patients have not responded favorably to non-surgical care, clinicians may also refer them for surgical consults (10 to 12%) [12,22,29,30].

While the best evidence for exercise-therapy and adjunct physical and rehabilitation medicine modalities have been recently synthesized [31,32], a systematic review of randomized control trials (RCTs) has not been completed for pharmaceutical or nutraceutical efficacy in over 10 years [33], and since then, new treatments have emerged (e.g., platelet rich plasma). Moreover, to our knowledge, a systematic synthesis of surgical treatment efficacy has not yet been undertaken [33]. Thus, we conducted a systematic review to answer the question of whether medical interventions (pharmaceutical, nutraceutical, or surgical) are efficacious in the management of PFP and PFOA, to inform primary care evidence-based decision making and identify knowledge gaps.

## 2. Methods

### 2.1. Study Design

We designed our review using the Preferred Reporting Items for Systematic Reviews and meta-analyses (PRISMA) guidelines [34]. We prospectively registered our study protocol as two separate protocols with the International Prospective Register of Systematic Reviews (PROSPERO) [35] (#CRD42017082527, #CRD42017078575). However, on account of the similarity of the research, study design and methods, and likelihood that a single paper would be most useful to clinicians, we subsequently combined them into a single study. To combine the two protocols, we adopted the methods from each protocol that provided the most rigorous and inclusive study design. We used the former protocol to select databases to search, resulting in five additional databases. We limited types of study designs to RCT only, in order to align with current best practices [26]. We included all types of control interventions, including wait-list or placebo. We defined the primary outcome as pain, though all outcomes from both protocols were still included. Finally, we removed the exclusion criterion of traumatic knee injury from the former protocol since this was no longer appropriate in regard to surgical studies.

### 2.2. Study Eligibility and Search Strategy

Our search strategy was developed by two co-authors (EMM, MvM) in collaboration with a reference librarian (Table 1). We included peer-reviewed RCTs investigating pharmaceutical, nutraceutical, or surgical interventions for PFP or PFOA. We included studies investigating PFP using intentionally broad criteria based on current recommended terminology [1]. Specifically, for PFP, we included individuals diagnosed either symptomatically or structurally, and included synonyms for PFP such as patellofemoral pain syndrome and chondromalacia patella, recognizing that terminology differs among different disciplines and has changed over time [1,36]. We also included individuals with PFOA because emerging evidence suggests that many conditions affecting the patellofemoral joint may lie along a pathological spectrum, with PFOA potentially representing a late stage of chronic PFP (though treatment approaches may differ) [7,8,16,17,18]. We included men and women of all ages. Diagnosis by any common clinical or structural criteria was eligible. We excluded studies targeting patellar instability. We included all types of control interventions (placebo, wait-list, other interventions). Our primary outcome was pain, and secondary outcomes included function, quality of life, structural outcomes on imaging, and biomechanics. We included studies reported in English, Dutch, French, or German.

Two investigators (EMM, HFH) independently screened titles and abstracts from seven databases: MEDLINE, PEDro, EMBASE, CINAHL, CENTRAL, SPORTDiscus, and Scopus. The first search included all publication dates from database inception to 12 December 2018, and a search update included publications to 4 May 2020. Relevant reference lists, including those of related review papers, were also screened. The same investigators independently screened all potentially eligible full-text publications. At both stages of screening, in cases of disagreement, a third investigator was consulted (CJB).

### 2.3. Risk of Bias

Two investigators (EMM, DT) independently evaluated the risk of bias (RoB) using Cochrane’s RoB v2.0 [37], and discrepancies were resolved with a third investigator (CJB). This tool reports on five domains of potential bias: randomization; deviations from intended interventions; missing outcome data; measurement of the outcome; and selection of the reported results. Each domain is summarized as having low or high bias, or some concerns. We contacted study authors in instances of missing information. Per Cochrane guidelines, a study was determined to be at overall high RoB if at least one domain was rated as having high RoB, or some concerns were identified across multiple domains. A study was considered at overall low RoB if all five domains had low RoB.

### 2.4. Data Extraction and Statistical Analyses

One author (EMM) extracted data including study population, participant characteristics, interventions, follow-up time points, and relevant outcomes. Data extraction was verified by a second author (CJB or DT). Where results were provided as legible figures, we digitized them and extracted outcomes (WebPlotDigitizer v.4.2, San Francisco, CA, USA). We contacted study authors where information was missing. We grouped follow-up times as: short-term (0 to 6 weeks), medium-term (7 weeks to 6 months), and long-term (greater than 6 months) [38]. Where adequate data were reported (or obtained from authors), we reported effect sizes as Hedge’s standardized mean differences (SMD) with 95% confidence intervals (CI) using Stata SE 15.1 (StataCorp LLC, College Station, TX, USA). SMDs enhance comparison across studies, particularly when different outcome measures are used to measure similar domains, by reporting effect sizes in units of pooled standard deviation. We conducted meta-analyses for any intervention with at least three trials of sufficient homogeneity and undertook best-evidence synthesis when this was not possible. We summarized the overall level of certainty of the evidence for each intervention using the Grading of Recommendation, Assessment, Development and Evaluation (GRADE) approach [39]. GRADE overall ratings range from a ”very low” to ”high” level of certainty, and are derived from consideration of five domains: RoB, consistency of effects, indirectness, imprecision, and publication bias. Using the GRADE approach represents a deviation from our original protocol that we undertook in order to better align with current best practices.

## 3. Results

We screened 6375 titles (Figure 1) and identified 22 eligible papers (20 studies). Meta-analysis was not possible due to study heterogeneity.

We rated fourteen studies as high RoB, four moderate, and two low (Figure 2). Only four studies registered trial protocols [40,41,42,43,44]. Eight studies did not report funding source [45,46,47,48,49,50,51,52], and eight were at least partly industry-funded [40,44,52,53,54,55,56,57].

### 3.1. Pharmaceutical or Nutraceutical Studies

In total, 14 publications (13 studies, 534 participants analyzed) investigated pharmaceutical or nutraceutical interventions [44,45,46,47,48,49,53,54,55,56,57,58,59,60]. One study enrolled participants with patellofemoral degeneration including OA [56], and the remaining studies mostly enrolled participants with PFP (Appendix A). Thirteen publications reported sex and age, approximately two/thirds of participants were women (*n* = 315), and the mean age was 28 years (Table 2). Only four studies reported body mass characteristics [44,47,48,60]. Three classes of interventions were rated as low certainty of evidence using GRADE, and seven were rated as very low certainty of evidence (Appendix A).

### 3.2. Oral Administration

#### 3.2.1. Nonsteroidal antiinflammatories, NSAIDs—Low Certainty of Evidence

One moderate [54] and two high RoB RCTs [49,55] investigated NSAIDs. While pain or global change improved with most interventions, aspirin (short-term) and naproxen (medium-term) were no different than placebo, and naproxen (short-term) was no different than diflunisal (Table 3). One other high RoB RCT found no within-arm change in pain (medium-term) using acetaminophen in a sample with patellofemoral degeneration [56].

#### 3.2.2. Chloroquine—Very Low Certainty of Evidence

One high RoB RCT of chloroquine compared to placebo reported medium-term improvement in ”spontaneous pain” [46], a combined measure of pain at rest, during walking and when first waking. There was inadequate detail to report SMDs.

#### 3.2.3. Glucosamine—Very Low Certainty of Evidence

One high RoB RCT compared oral glucosamine to NSAID (acetaminophen) in patellofemoral degeneration [56]. Medium-term results favored glucosamine in pain (SMD 2.5 [95% CI 1.1, 3.9]) and function outcomes (American Knee Society score, SMD 4.1 [2.2, 6.0]).

### 3.3. Injections

#### 3.3.1. Glycosaminoglycan Polysulphate (GAGPS)—Very Low Certainty of Evidence

Two high RoB RCTs (three publications) investigated GAGPS injections [48,58,59]. Five intra-articular injections were no better than placebo for pain and function for up to five years after treatment [58,59]. Twelve intra-muscular injections were no better than placebo at short- and medium-term for pain, function, and physician-determined overall effect. However, long-term results (one year) favored GAGPS injections for pain descending stairs (SMD 1.6 [0.8, 2.5]), pain squatting (SMD 1.3 [0.5, 2.2]), hindrance in sport (SMD 0.8 [0.1, 1.6]), and physician’s evaluation of being ”improved” (relative risk, RR 3.6 [1.2, 10.4]).

#### 3.3.2. Hyaluronic Acid—Low Certainty of Evidence

One low RoB RCT compared a single intra-articular injection of hyaluronic acid to a sham injection [44]. Investigators reported no clinically meaningful between-group differences for up to six months of follow-up. Our SMD calculations, based on complete case analysis, showed no differences in pain, Tegner scores, or strength, but the group receiving hyaluronic acid had worse function (Knee injury and Osteoarthritis Outcome Score [KOOS], SMD −0.6 [−1.0,−0.2]; Anterior Knee Pain Scale [AKPS], SMD-0.5 [−1.0, −0.1].

#### 3.3.3. Platelet Rich Plasma (PRP)—Very Low Certainty of Evidence

One high RoB RCT compared three intra-articular PRP injections to one injection [47]. Both arms demonstrated medium-term within-group improvements in self-reported pain/function, balance, coordination, and endurance. However, only pain/function showed a between-group difference favoring three injections over one (AKPS SMD 1.1 [0.3, 2.0]).

#### 3.3.4. Botox—Low Certainty of Evidence

One moderate RoB RCT evaluated Botox injected into vastus lateralis compared to placebo [57]. Medium-term results favored Botox for self-reported pain/function (AKPS SMD 0.9 [0.1, 1.8]), and pain in squatting (SMD 1.1 [0.2, 1.9]) and kneeling (SMD 1.4 [0.5, 2.3]), but no difference with pain in stairs or walking.

#### 3.3.5. Anabolic Steroids—Very Low Certainty of Evidence

One high RoB RCT compared nandrolone phenylpropionate injection to placebo [53]. Short-term results favored the treatment arm in terms of the number of participants becoming moderately improved or resolved (RR 8.7 [2.3, 32.7]).

### 3.4. Transdermal

#### 3.4.1. NSAIDs (Phonophoresis)—Very Low Certainty of Evidence

One moderate RoB RCT compared olive oil (phonophoresis) to NSAID (phonophoresis) and placebo (ultrasound alone) [60]. At one week, within-arm improvements were seen for olive oil (pain and function) and piroxicam (function). By two weeks, all three arms improved (pain and function), with olive oil improving more than placebo (function) but not more than piroxicam (values not reported), and piroxicam no different than placebo.

#### 3.4.2. Corticosteroid (Iontophoresis or Phonophoresis)—Very Low Certainty of Evidence

One high RoB RCT compared Hexadrol with topical Xylocaine (phonophoresis), Hexadrol with topical Xylocaine (iontophoresis), ice, and ultrasound with ice [45]. Short term, more participants reported subjective improvement using ultrasound with ice than either corticosteroid arm, though this does not appear to be significant (inadequate reporting).

#### 3.5. Surgical Studies

Eight publications (seven studies, 411 participants analyzed) investigated surgical interventions [40,41,42,43,50,51,52,61]. The source population was more heterogeneous compared to pharmaceutical and nutraceutical studies (Table 2, Appendix A): two studies enrolled participants with isolated PFOA [40,43], one with isolated patellar cartilage lesions [51], and the remaining mostly PFP. Approximately 72% of participants were women (*n* = 294), with a mean age of 41 years (mean age range 28 to 64). Only two studies reported body mass characteristics [41,42,43]. Four classes of interventions were rated as very low certainty of evidence using GRADE, and one was rated as moderate certainty of evidence (Appendix A).

### 3.6. Arthroscopy

#### 3.6.1. Standard Arthroscopy—Very Low Certainty of Evidence

One high RoB RCT (two publications) compared arthroscopy plus exercise to exercise alone [41,42]. Specific procedures were as indicated at surgery, (e.g., plica resection, abrasion of chondral lesions, shaving of synovium). At medium- and long-term (two and five year) follow-ups, surgery was no better than exercise alone in pain or function (Table 3).

#### 3.6.2. Radiofrequency Debridement—Very Low Certainty of Evidence

One high RoB RCT compared arthroscopy with a bipolar radiofrequency probe compared to arthroscopy with a standard mechanical shaver for treating isolated patellar cartilage lesions [51]. Long-term results favored the radiofrequency probe in terms of self-reported pain/function (SMD 1.7 [0.9, 2.4]).

### 3.7. Open Surgeries

#### 3.7.1. Lateral Retinacular Surgery—Very Low Certainty of Evidence

One high RoB RCT compared open lateral retinacular lengthening to open lateral retinacular release [50]. There was no difference three months after surgery, but at six months, results favored lengthening in terms of less quadriceps atrophy (SMD 1.2 [0.4, 2.0]). Long-term results favored lengthening for quadriceps atrophy (SMD 1.6 [0.7, 2.5]) and pain/function (AKPS SMD 0.8 [0.1, 1.6]).

One high RoB RCT compared open lateral retinacular lengthening to arthroscopic lateral retinacular release [61]. Long-term results (up to six years) favored open lengthening for pain/function (modified Lysholm, inadequate detail to report values) but no difference for return to sport level or strength.

#### 3.7.2. Anterior Tibial Tuberosity Displacement—Very Low Certainty of Evidence

One high RoB RCT compared anterior tibial tuberosity displacement plus debridement to debridement alone [52]. Long-term results favored anterior tibial displacement in terms of the surgeon-determined proportion rated as good or excellent (RR 2.6 [1.2, 5.4]).

#### 3.7.3. Patellofemoral Arthroplasty—Moderate Certainty of Evidence

Two RCTs (one low and one moderate RoB) compared patellofemoral arthroplasty to total knee arthroplasty in isolated PFOA [40,43]. In one study [40], short- and medium-term results favored patellofemoral arthroplasty for two pain subscales (KOOS Symptoms, SMD 0.6 [0.2, 1.0]; and SF-36 Bodily Pain, SMD 0.5 [0.1, 0.9]), but results did not differ for remaining KOOS and SF-36 subscales. In the same study, long-term results (2 years post-surgical) only remained in favor of patellofemoral arthroplasty for KOOS Symptoms (SMD 0.5 [0.1, 0.9]). The other study reported no differences in any outcome at 1 year after surgery, including several outcomes measured five years after surgery [43].

## 4. Discussion

Medical interventions are common for individuals with PFP or PFOA [12,22,27,29,30]. Given the high prevalence and potential for chronicity in these patellofemoral conditions, it is clinically imperative to make evidence-informed decisions when managing these patients. Our results suggest that most pharmaceutical, nutraceutical, or surgical interventions for PFP and PFOA have not undergone sufficiently rigorous investigation to warrant their use. Rather than placebo, usual care, or a control of known efficacy, many interventions were compared to other interventions that have not yet themselves undergone rigorous evaluation. Such studies add little to the overall evidence. Most studies included in this review were of high RoB, and results should thus be interpreted with caution. Replication studies with lower RoB would likely reduce the number of positive results in this review and shift larger effect sizes towards the null [62]. Studies with negative findings reported in this review, however, are less likely to change meaningfully in the presence of less bias [62].

There is almost no robust evidence regarding pharmaceutical or nutraceutical interventions for PFP or PFOA. Highlighting interventions more commonly prescribed in general practice: short-term, oral NSAIDs were no better than placebo (low certainty of evidence); medium-term, large improvements were reported with oral glucosamine compared to NSAID in a sample with patellofemoral degeneration including OA (very low certainty of evidence); and long-term, large improvements were reported with intra-muscular GAGPS injections compared to placebo (very low certainty of evidence). Regarding GAGPS, the difference in results between the two studies could be due to the different delivery site (intra-articular vs. intra-muscular), dose-response (five injections in the study with negative findings vs. 12 in the study favoring GAGPS), or other factors such as sample selection (different proportion of women, for example). Having said this, long-term results were only measured in one of the two studies, and it is unclear how to explain why results would differ at one year but not at earlier time points.

There is very low to moderate certainty of evidence regarding surgical interventions. Highlighting more common surgical interventions: standard arthroscopy plus exercise was no better than exercise alone for PFP. The only intervention in the present study with moderate certainty of evidence was for patellofemoral arthroplasty. This evidence suggests possible early benefits in some symptoms compared to total knee arthroplasty for treating isolated PFOA, but likely no differences between interventions one to five years following surgery, suggesting that both approaches may be equally effective during this period of follow-up. One of these studies was rated as low RoB, and the other study introduced some bias through its pragmatic design, with surgeons deviating from protocol in several cases based on clinical judgement. Rates of follow-up procedures were similar between groups in one study, and higher in the total knee group in the other study. Additional benefits of patellofemoral arthroplasty may include it being a less invasive procedure with quicker post-operative recovery and preservation of bone stock [63]. While longer follow-ups are still required, as are studies comparing patellofemoral arthroplasty to non-surgical interventions, patellofemoral arthroplasty appears to be as effective as total knee arthroplasty for treating isolated PFOA, and may thus be a valid surgical option [29,30,64].

Pharmaceutical, nutraceutical, and surgical interventions for the treatment of PFP and PFOA are generally administered on the premise that they address underlying inflammation or structural abnormalities that cause the pain [12,22]. However, the source of pain is highly debated [1], and much like in knee OA [65], pain and structure are not strongly correlated [66,67], and pain is highly variable and subjective. This may explain why some interventions included in this review offer no benefit over placebo. It is possible that a subgroup of individuals with PFP or PFOA may represent ”responders” to disease-modifying treatment approaches, and future research may be able to identify these individuals and target treatment appropriately. On the other hand, a growing body of literature suggests that moving beyond localized physical interventions and treating individuals more holistically may hold promise for better outcomes [68]. Examples of such interventions may include improving patient education, weight management, treatments targeting pain perception and neuroplasticity, or psychological support [63]. Non-physical approaches such as these will likely be most effective in combination with an exercise-based approach, which is recommended as a core treatment for PFP and PFOA [23,24,25,26].

Limitations to the present study include broad eligibility criteria, which resulted in study heterogeneity. For example, we included both structural diagnoses and clinical definitions of PFP, including PFOA. However, these diagnoses fall along a spectrum of patellofemoral disease, so presenting interventions in a single review provides evidence across this spectrum. In addition, we included any pharmaceutical, nutraceutical, or surgical intervention, and did not limit control interventions. These broad inclusion criteria enabled us to capture more studies relating to our research question but should be considered in the context of synthesizing and interpreting of results, with very few studies including a control reflecting treatments of efficacy, usual care, or wait and see. Sample sizes were generally small in our included studies (11 studies had fewer than 50 participants). It was not possible to formally evaluate publication bias on account of the limited number of studies published for each interventions [69]. Finally, while limiting our inclusion to RCTs is arguably a strength of our study, one of the possible limitations is that we may have missed observational studies that reported poor or even harmful outcomes of some interventions. Thus, further evaluation of some procedures, such as anterior tibial tuberosity displacement, may be both unnecessary and inappropriate [70].

The results of the present systematic review suggest that clinical decisions regarding pharmaceutical, nutraceutical, and surgical interventions are currently being made with very little supporting evidence. Limited efficacy of these treatments in the target population(s) should be considered within the broader context of therapeutic options for PFP and PFOA. In particular, therapeutic exercise consistently demonstrates large effect sizes on pain and function in short and long-term studies [71], and evidence continues to emerge to guide exercise types and dose for improving these outcomes [32,72,73]. The present review suggests that rigorous research is urgently needed to support current best-practices in primary care for the management of PFP and PFOA. This is particularly relevant given the chronic nature of PFP, its association with PFOA [7,8,14,15,16,17,18], and the likelihood of PFOA progressing further towards whole knee OA [21].

In conclusion, our systematic review found very limited to moderate certainty of evidence, primarily based on single studies, and often comparing interventions to other interventions with no established efficacy. This limits our ability to offer robust recommendations as to which medical interventions may be effective in the management of PFP and PFOA.

## Figures and Tables

**Figure 1 jcm-09-03397-f001:**
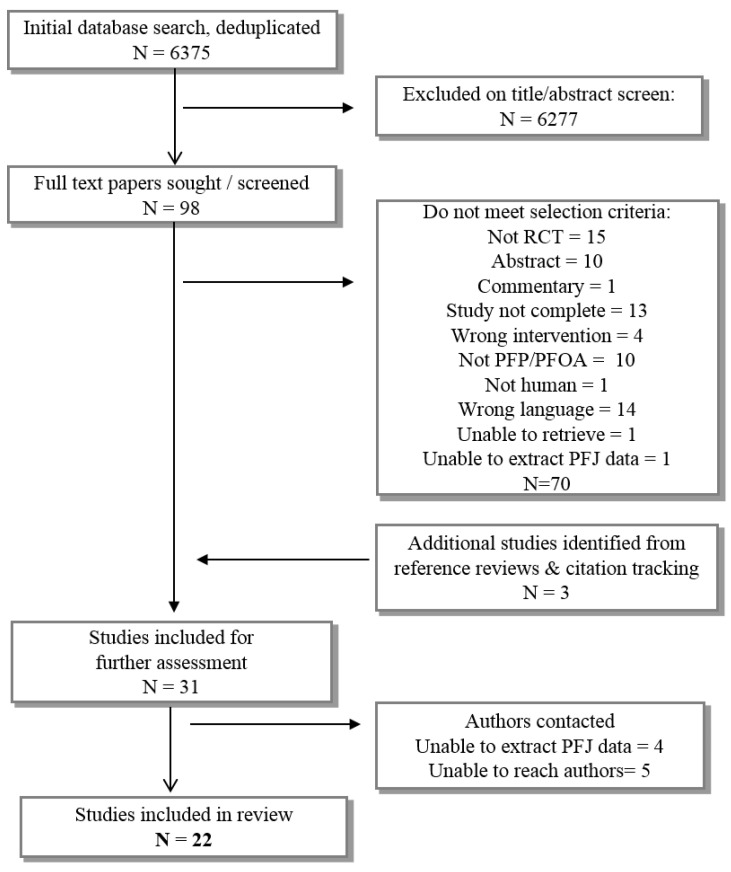
Flow chart of study selection.

**Figure 2 jcm-09-03397-f002:**
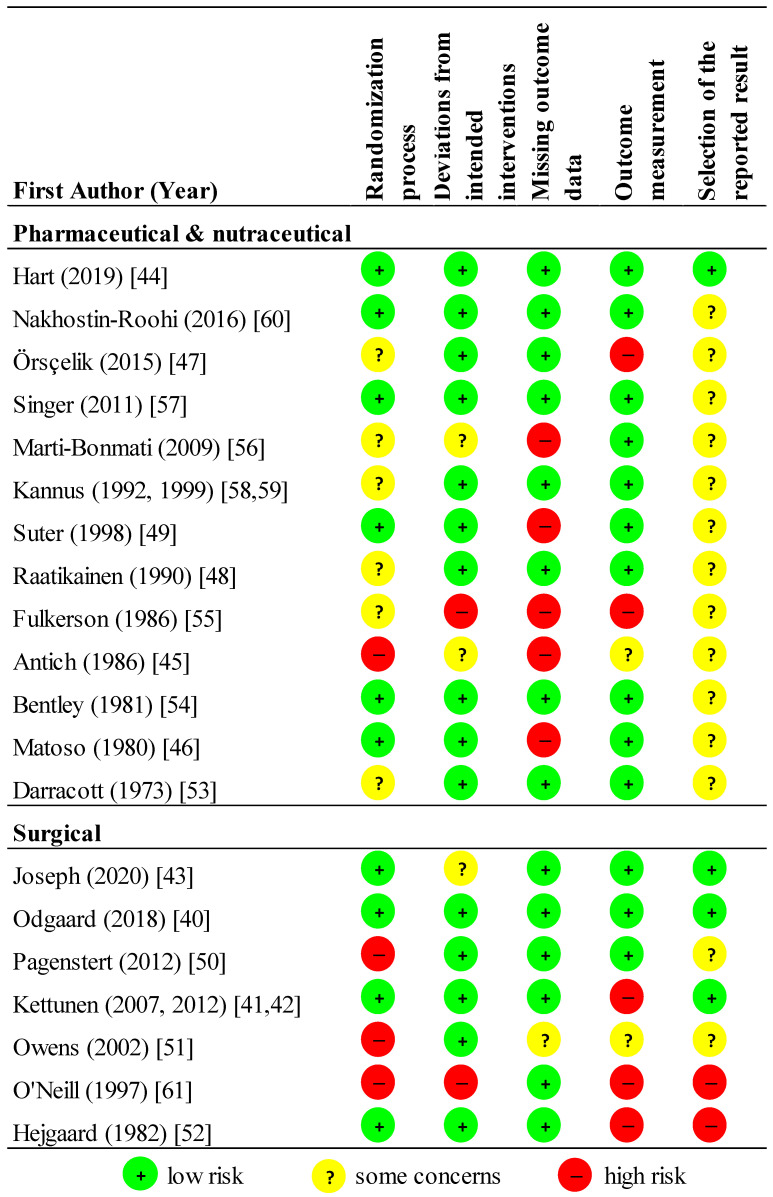
Risk of bias (RoB) summary. Where more than one manuscript of the same study is published, they are reported together.

**Table 1 jcm-09-03397-t001:** Search string for Medline (Ovid).

(Patellofemoral Pain Syndrome/OR Chondromalacia Patellae/OR ((Patellofemoral Joint/OR patella/) AND (osteoarthritis/OR osteoarthritis, knee/OR pain/OR Joint Diseases/OR Cartilage Diseases/)) OR Chondromalacia Patellae/OR (((femoropatell* OR patell* OR retropatell*) ADJ6 (pain* OR osteoarthrit* OR osteo-arthrit* OR arthralgi* OR syndrome* OR Chondromalac* OR dysfunction* OR chondropath*)) OR ((femoropatell* OR patell* OR retropatell*) ADJ6 (degenerat*) ADJ6 (arthrit* OR cartilag*)) OR (anter* ADJ3 knee ADJ3 pain*) OR ((lateral* OR odd) ADJ3 (compress* OR facet OR pressure*) ADJ3 syndrome*)).ab,ti,kw) AND (Exp Controlled clinical trial/OR “Double-Blind Method”/OR “Single-Blind Method”/OR “Random Allocation”/OR (random* OR factorial* OR crossover* OR cross over* OR placebo* OR ((doubl* OR singl*) ADJ blind*) OR assign* OR allocat* OR volunteer* OR trial OR groups).ab,ti.) NOT (Animals/NOT Humans/)

NB Syntax was developed in MEDLINE using keywords and medical subject headings, then adapted for PEDro, EMBASE, CINAHL, CENTRAL, SPORTDiscus, and Scopus using their respective indexing vocabularies.

**Table 2 jcm-09-03397-t002:** Participant characteristics.

First Author Year	Sample Randomized *n*	Sample Completed *n*		Women % (*n*)		Age Mean (SD) y		BMI Mean (SD) kg/m^2^		Height Mean (SD) cm		Weight Mean (SD) kg	
Pharmaceutical and Nutraceutical		Int	Cont	Int	Cont	Int	Cont	Int	Cont	Int	Cont	Int	Cont
Hart 2019 [44]	86	42	38	76% (34)	76% (31)	26.0 (7.0)	28.1 (8.4)	26.4 (5.3)	25.8 (5.1)	168.9 (9.2)	168.3 (8.6)	75.7 (16.6)	73.5 (17.6)
Nakhostin-Roohi 2016 [60]	93	31	31 * 31 *	100% (93)		26.0 (8.9)	26.2 (10.0) 27.2 (9.5)	NR	NR NR	160.7 (11.1)	164.2 (7.2) 1.6 (8.9)	59.3 (9.5)	58.2 (11.1) 61.0 (12.8)
Örsçelik 2015 [47]	30	10	20	40% (4)	35% (7)	28.7 (6.0)	27.2 (5.7)	24.2 (3.4)	24.0 (3.0)	170.8 (6.6)	174.5 (7.9)	71.1 (12.5)	73.4 (12.8)
Singer 2011 [57]	24	14	10	57% (8)	90% (9)	31.5 (r15,48)	27.4 (r18,44)	NR	NR	NR	NR	NR	NR
Marti-Bonmati 2009 [56]	20	10	6	63% (10)		39 (18)		NR	NR	NR	NR	NR	NR
Kannus 1992 [58] and	5	16	16 ^†^ 17 ^†^	53% (28)		27 (9)		NR	NR NR	NR	NR NR	NR	NR NR
1999 [59]	As above	45		aa		aa		aa		aa		aa	
Suter 1998 [49]	42	19	17	31% (13)		35.6 (8.4)		NR	NR	NR	NR	NR	NR
Raitikainen 1990 [48]	31	14	15	29% (4)	20% (3)	29.1 (7.7)	30.2 (6.6)	24.6 (3.3)	25.1 (3.2)	173.6 (9.0)	177.0 (8.8)	74.8 (14.0)	78.9 (13.9)
Fulkerson 1986 [55]	56	20	16	78% (28)		32.3 (??)		NR	NR	NR	NR	NR	NR
Antich 1986 [45]	64 (86 k)	9 k	21 k ^‡^ 13 k ^‡^ 16 k ^‡^	NR		NR		NR	NR	NR	NR	NR	NR
Bentley 1981 [54]	30	16	13	72% (21)		25 (??)		NR	NR	NR	NR	NR	NR
Matoso 1980 [46]	33	12	10	53% (10)	64% (9)	28 (r16,50)	30 (r16,46)	NR	NR	NR	NR	NR	NR
Darracott 1973 [53]	43	23	20	0% (0)	15% (3)	22.9 (4.4)	20.8 (3.3)	NR	NR	NR	NR	NR	NR
**Surgical**													
Joseph 2020 [43]	64	31	29	71% (22)	90% (26)	64.7 (10.5)	64.4 (12.8)	28.9 (6.7)	29.2 (4.2)	NR	NR	NR	NR
Odgaard 2018 [40]	100	46	47	77% (77)		64 (8.9)		NR		NR		NR	
Pagenstert 2012 [50]	(28)	14	14	79% (11)	71% (10)	47.6 (9.9)	48.0 (11.6)	NR	NR	NR	NR	NR	NR
Kettunen 2007 [41]	56	27	25	61% (17)	64% (18)	28.4 (7.5)	28.4 (5.6)	24.1 (3.3)	23.8 (3.6)	171.7 (10.2)	172.4 (9.6)	69.0 (19.3)	71.4 (15.1)
2012 [42]	As above	24	20	aa	aa	aa	aa	aa	aa	aa	aa	aa	aa
Owens 2002 [51]	48	20	19	100% (20)	100% (19)	36.9 (r 30,45)	37.5 (r 30,45)	NR	NR	NR	NR	NR	NR
O’Neill 1997 [61]	91	43	43	60% (26)	58% (25)	27.2 M 18 (r13,56) W 33 (r15,47)	28.7 M 22 (r13,33) W 34 (r14,59)	NR	NR	NR	NR	NR	NR
Hejgaard 1982 [52]	42	20	22	50% (10)	59% (13)	28 (r18,38)	32 (r19,50)	NR	NR	NR	NR	NR	NR

BMI = body mass index; k = knees; NR = not reported; Int = intervention; Cont = control; aa = as above; iPFOA = isolated patellofemoral osteoarthritis. * cont top = piroxicam, cont bottom = base gel. ^†^ cont top = exercise alone, cont bottom = placebo injection. ^‡^ cont top = iontophoresis; cont mid = US/ice; cont bottom = ice.

**Table 3 jcm-09-03397-t003:** Standardized mean differences (SMD) and relative risks (RR) of all included studies (SMD > 0 and RR > 1 indicate improvement of intervention group relative to control).

First Author Year	Intervention	Control	Outcome	Short Term * <6 Weeks	Medium Term 7 Weeks–6 Months	Long Term, >6 Months	Adverse Events
Pharmaceutical and Nutraceutical				SMD (95% CI)	SMD (95% CI)	SMD (95% CI)	
Hart 2019 [44]	Hyaluronic acid IA inj	Placebo	VAS single leg squat	−0.3 (−0.7, 0.2)	−0.3 (−0.7, 0.2)		NR
			KOOS	**−0.9 (−1.3, −0.4)**	**−0.6 (−1.0,−0.2)**		
			AKPS	**−0.6 (−1.1, −0.2)**	**−0.5 (−1.0,−0.1)**		
			Tegner	−0.3 (−0.7, 0.1)	−0.2 (−0.6, 0.3)		
			Knee extension	0.0 (−0.4, 0.4)	−0.1 (−0.6, 0.3)		
			Knee ratio	0.0 (−0.4, 0.4)	−0.3 (−0.7, 0.2)		
Nakhostin-Roohi 2016 [60]	Olive oil (phonophoresis)	Piroxicam (phonophoresis)	WOMAC Pain	NR			NR
			WOMAC Stiffness	NR			
			WOMAC Physical Function	NR			
Örsçelik 2015 [47]	PRP, 3 IA inj	PRP, 1 inj	AKPS		**1.1 (0.3, 2.0)**		Some localized pain for several weeks after injections (numbers/arm not reported)
			Quadriceps peak torque		0.1 (−0.7, 0.8)		
Singer 2011 [57]	Botox IM inj	Placebo	AKPS		**0.9 (0.1, 1.8)**		6 (43%) slight distal thigh asymmetry; few (numbers and arm NR): temporary bruising or pain last up to several days.
			Pain stairs		0.3 (−0.5, 1.1)		
			Pain squat		**1.1 (0.2, 1.9)**		
			Pain kneel		**1.4 (0.5, 2.3)**		
			Pain walk		−0.9 (−1.7, 0.0)		
Marti-Bonmati 2009 [56]	Glucosamine	Acetaminophen	AKSS		**2.5 (1.1, 3.9)**		NR
			VAS (task not specified)		**4.1 (2.2, 6.0)**		
Kannus 1992 and 1999 [58,59]	GAGPS IA inj	Placebo	No pain, 1 leg jump	RR 0.9 (0.4, 1.8)	RR 0.8 (0.5, 1.3)	NR	0 intervention vs. 1 control reactive synovitis to injection (discontinued treatment)
			No pain, 25 squats	RR 1.1 (0.7, 1.6)	RR 1.1 (0.7, 1.6)		
			Excellent rating	RR 0.9 (0.5, 1.8)	RR 1.4 (0.8, 2.4)		
			Return full activity	RR 1.4 (0.8, 2.4)	RR 1.4 (0.9, 2.0)		
			Lysholm	0.2 (−0.5,0.8)	0.2 (−0.5, 0.9)		
			Tegner	0.4 (−0.3, 1.1)	−0.1 (−0.6, 0.8)		
			Pain with activity	0.2 (−0.9, 0.5)	0.0 (−0.7, 0.7)		
Suter 1998 [49]	Naproxen	Placebo	Pain resisted extension	0.5 (−0.2, 1.1)			NR
Raatikainen 1990 [48]	GAGPS IM inj	Placebo	Pain down stairs	−0.1 (−0.6, 0.9)	0.3 (−0.4, 1.1)	**1.6 (0.8, 2.5)**	1 intervention, sweating and dizziness after 4th injection, resolved and continued
			Pain squat	0.3 (−0.4, 1.1)	0.3 (−0.4, 1.1)	**1.3 (0.5, 2.2)**	
			Physician ”Improved”	RR 1.4 (0.6, 3.4)	RR 1.7 (0.5, 6.1)	**RR 3.6 (1.2, 10.4)**	
			Hindrance—sport	0.4 (−0.3, 1.2)	0.5 (−0.3, 1.2)	**0.8 (0.1, 1.6)**	
			Hindrance—life	−0.2 (−0.9, 0.6)	0.3 (−0.4, 1.1)	0.7 (−0.0, 1.5)	
Fulkerson 1986 [55]	Diflunisal	Naproxen	Significant pain relief	RR 0.9 (0.5, 1.5)			11 (55%) diflunisal (headache, gastric distress, etc.) vs. 7 (44%) naproxen (drowsiness, headache, etc.)
Antich 1986 [45]	Hexadrol (phonophoresis)	Ultrasound plus ice	Subjective improvement	[RR 0.7 (0.2, 2.2)] ^†^			NR
Bentley 1981 [54]	Palaprin	Placebo	Global improvement	RR 1.2 (0.4, 3.4)			NR
Matoso 1980 [46]	Chloroquine	Placebo	Spontaneous pain		NR		15 (79%) intervention (6 discontinued treatment) vs. 4 (29%) controls: visual trouble, digestive trouble, headache, vertigo
Darracott 1973 [53]	Nandrolone phenylpropionate inj	Placebo	Improved	**RR 8.7 (2.3, 32.7)**			**NR**
**Surgical**							
Joseph 2020 [43]	Patellofemoral arthroplasty	Total knee arthroplasty	WOMAC (1y):				Superficial infections: 4 PFA, 5 TKA—all treated with antibiotics. 4 TKA required further interventions: 1 arthroscopic facetectomy, 1 manipulation under anesthesia, 2 aspiration/steroid injection
			Pain			−0.3 (−0.8, 0.2)	
			Stiffness			−0.2 (−0.7, 0.3)	
			Function			−0.1 (−0.6, 0.4)	
			AKSS (1y):				
			Knee			0.1 (−0.5, 0.6)	
			Function			−0.2 (−0.4, 0.7)	
			UCLA (1y)			0.3 (−0.2, 0.8)	
			EQ5D3L (5y)			0.2 (−0.4, 0.8)	
			OKS (5y)			−0.1 (−0.8, 0.5)	
			Pain-free years (5y)			−0.6 (−1.2, 0.1)	
			Satisfied (5y)			RR 0.9 (0.6, 1.2)	
Odgaard 2018 [40]	Patellofemoral arthroplasty	Total knee arthroplasty	KOOS:				Intervention: 2 deaths unrelated to surgery, 2 revisions (1 of trochlear component, 1 to TKA), 2 other surgical procedures; control: 5 other surgical procedures
			Pain	0.4 (−0.0, 0.8) ^‡^	0.4 (−0.0, 0.8)	0.1 (−0.3, 0.5)	
			Symptoms	**0.6 (0.2, 1.0)**	**0.8 (0.4, 1.2)**	**0.5 (0.1, 0.9)**	
			ADL	0.2 (−0.2, 0.6)	0.1 (−0.3, 0.5)	−0.1 (−0.6, 0.3)	
			SPR	0.3 (−0.1, 0.7)	0.3 (−0.1, 0.7)	0.3 (−0.2, 0.7)	
			QOL	0.0 (−0.4, 0.4)	0.3 (−0.1, 0.7)	0.1 (−0.4, 0.5)	
			SF-36 Bodily Pain	**0.5 (0.1, 0.9)**	**0.4 (0.0, 0.8)**	0.3 (−0.1, 0.7)	
Pagenstert 2012 [50]	Open lateral retinacular lengthening	Open lateral retinacular release	AKPS		0.5 (−0.2, 1.3)	**0.8 (0.1, 1.6)**	No surgical complications. At 2 years, recurrence of symptoms in 1 lengthening and 2 release; and over-release in 5 release arm
			Quadriceps atrophy		**1.2 (0.4, 2.0)**	**1.6 (0.7, 2.5)**	
Kettunen 2007 and 2012 [41,42]	Arthroscopy plus exercise	Exercise alone	AKPS			−0.0 (−0.6, 0.5)	Intervention: 1 delayed exercise due to pain, 1 refused exercise after surgery; 3 controls had arthroscopic surgery
			Pain down stairs			−0.1 (−0.7, 0.4)	
			Pain up stairs			−0.0 (−0.5, 0.5)	
			Pain sit to stand			0.2 (−0.3, 0.7)	
Owens 2002 [51]	Debridement, radiofrequency probe	Debridement, standard shaver	Fulkerson-Shea			**1.7 (0.9, 2.4)**	NR
O’Neill 1997 [61]	Open lateral retinacular lengthening	Arthroscopic lateral retinacular release	Tegner			RR 1.1 (1.0, 1.2)	Intervention: 1 stitch abscess; 1 skin incision keloid; 1 unable to achieve 90° knee flexion; control: 1 iliotibial band contracture, 1 hematoma, 1 superficial infection, 1 hematoma with infection after self-draining
			Quadriceps open chain strength deficit			RR 1.1 (0.8, 1.6)	
			Quadriceps atrophy			RR 1.2 (0.9, 1.5)	
			Knee score (modified Lysholm)			**NR**	
Hejgaard 1982 [52]	Anterior displacement of tibial tuberosity plus debridement	Debridement	Surgeon, Good to Excellent			**RR 2.6 (1.2, 5.4)**	Intervention: 2 (10%) effusion, 2 (10%) thromboembolism; control: 5 effusion (23%)

Interventions for pharmaceutical/nutraceutical are taken orally unless otherwise stated. Outcomes reported are for pain, function, and global changes only—other outcomes (e.g., balance, biomechanics) were reported so rarely that comparisons are not possible. * Where multiple evaluations occur within a given follow-up time category, we report only the longest follow-up within that time period. ^†^ Approximate estimate only—numbers reported are knee-level comparisons, not person-level, with no indication of how many participants had both knees assessed. ^‡^ Values were extracted from figures; however, we assumed that the error bars represent 95% CI and NOT standard error of the mean as is reported in the publication figures—this is more congruent with the publication’s reported results. SMD = standardized mean difference; RR = relative risk; NR = not reported or inadequate detail to extract data; Int = intervention arm; GAGPS = glycosaminoglycan polysulphate; PRP = platelet rich plasma; inj = injection; IA = intra-articular injection; IM = intra-muscular injection; SF-36 = Medical Outcomes Score short form 36; KOOS = Knee injury and Osteoarthritis Outcome Score; ADL= activities of daily living; SPR = sports and recreation; QOL = quality of life; AKPS = anterior knee pain scale; WOMAC = Western Ontario McMaster Osteoarthritis Index; AKSS = American Knee Society Score; UCLA = University of California Los Angeles Physical Activity Questionnaire; EQ5D3L = EuroQol five-dimension, three-level questionnaire; OKS: Oxford Knee Score. Bold indicates statistical significance (*p* < 0.05).

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
