# Peer review of "Medical Interventions for Patellofemoral Pain and Patellofemoral Osteoarthritis: A Systematic Review"

_jcm, 2020, doi:10.3390/jcm9113397_

Round 1

Reviewer 1 Report

The authors have addressed some of my concerns. However, important issues remain:

Please include observational studies, this would greatly improve the quality of your review especially since there are presently not enough studies to evaluate publication bias.

A recent study from Cochrane has shown that estimates from RCTs and observational studies are, most of the time, not that different and that it is important to focus on other study level factors than design alone:

“Our results underscore that it is important for review authors to consider not only study design, but the level of heterogeneity in meta-analyses of RCTs or observational studies. A better understanding of how these factors influence study effects might yield estimates reflective of true effectiveness.”

From :

Cochrane Database Syst Rev . 2014 Apr 29;(4):MR000034. doi: 10.1002/14651858.MR000034.pub2.

Healthcare outcomes assessed with observational study designs compared with those assessed in randomized trials

Andrew Anglemyer , Hacsi T Horvath, Lisa Bero

Furthermore the 2 protocols (#CRD42017082527, #CRD42017078575) are not “with highly overlapping study design and methods” as the authors suggest in their rebuttal. For instance: #CRD42017078575 says:

“A wide range of methodological designs have been used in assessing the efficacy of surgery in PFP and PFOA. Due to the small number of randomised controlled trials available, studies of both randomised and non-randomised designs will be included.”

And later:

“Publications comparing surgery to the following interventions will be included:

  1. Surgical intervention (another form of surgery);
  2. Non-surgical intervention (physiotherapy, exercise, taping, bracing etc.);
  3. Pharmacological intervention (injection therapy, anti-inflammatories etc.);
  4. No intervention (wait and see).

Publications with no comparison groups will still be eligible for inclusion in this review.”

It seems that all observational studies and publications with no comparison groups were dropped. This is not mentioned nor discussed in the paper.

This is exactly why it is better not to combine the two reviews: they are very different both in design and in research questions.

Reviewer 2 Report

The authors have revised their manuscript in an appropiate way following all the suggestions of the Reviewers. Therefore, I think that the article must be accepted for publication. 

Author Response

Response: Thank you very much for your time and helpful feedback.

Reviewer 3 Report

Thank you for your revised manuscript.

I checked all revisions that was suggested by other reviewers.

I agree with your edition and admit the limitation of this topic.

Once again, Thank you for your effort.

Author Response

(The authors gave the same response as above.)

Round 2

Reviewer 1 Report

n

This manuscript is a resubmission of an earlier submission. The following is a list of the peer review reports and author responses from that submission.

Round 1

Reviewer 1 Report

I want to congratulate the authors on a wonderful study which is required for field. The data is comprehensive and well detailed and raises important considerations on the impact of different interventions. This type of study and summary was lacking and is excellent to have now. 

Author Response

Reviewer 1

Point 1.1: I want to congratulate the authors on a wonderful study which is required for field. The data is comprehensive and well detailed and raises important considerations on the impact of different interventions. This type of study and summary was lacking and is excellent to have now. 

Response 1.1: We thank Reviewer 1 very much for their time and positive feedback.

Reviewer 2 Report

The authors have performed a systematic review on pharmaceutical, nutraceutical and surgical interventions for patellofemoral pain and patellofemoral osteoarthritis for which they included 22 RCTs. No meta-analyses was performed.

The authors conclude that the above mentioned interventions are being performed with little supporting evidence. This claim is made by only evaluating RCTs. Can the authors really claim no or little evidence when they themselves have neglected all observational studies? The review could and should be improved by including observational studies.

The Van Tulder criteria for level of evidence are not accurately used. Table 6 from the original Van Tulder paper:

Table 6. Levels of Evidence

Strong—consistent findings among multiple high quality RCTs*

Moderate—consistent findings among multiple low quality RCTs and/orCCTs and/or one high quality RCTLimited—one low quality RCT and/or CCT

Conflicting—inconsistent findings among multiple trials (RCTs and/orCCTs)No evidence from trials—no RCTs or CCTs

Consistency and quality should be clearly defined a priori

* There is consensus among the Editorial Board of the BRG that strong evidence can only be provided by multiple high quality trials that replicate findings of other researchers in other settings.

CCT means controlled clinical trial which are non-randomized observational studies.

Also regarding the levels of evidence, the authors should consider using the GRADE system, which is more modern than the Van Tulder (2003) system.

The authors have combined two reviews (#CRD42017082527, #CRD4201707857). They have not explained why it was necessary to combine them and the practice of combining two protocols raises some eyebrows. The whole point of an a priori-protocol is to allow readers and reviewers to compare what was done with what was planned. Combining two protocols does not allow that anymore. It would have been better to register a new protocol and close the old ones in Prospero.

Is Prospero aware that you have combined the two protocols?

The way the results (both tables and text) are presented is cumbersome, hard to read and difficult to digest. Please organise your results in a reader friendly manner.

The citations from the tables and figures do not correspond with the references at the end of the paper. For instance: Table 4 Darracott [45] is reference 48 in the list. Please check and correct all citations and references.

There are concerns regarding the selection procedure. Darracott (1973) [45] (or 48), is a “trial” and it is not mentioned in the that paper if the patients were randomized and given that it was published in 1973 a trial could mean a comparative study (e.g. pseudo randomisation).

Publication bias was not assessed.

Author Response

/

Reviewer 3 Report

The systematic review is very interesting and well done. However I recommend the following changes to the authors to improve the quality of their paper:

* CHANGE THE TITLE: My suggestions is "Non-surgical and surgical treatments for patellofemoral pain and patellofemoral osteoarthritis: A systematic review.

* Delete the term "nutraceutical" all along the paper, tables and figures (it is misleading).

* In Figure 1, Table 2, Table 3 and Table 4, put the references in chronological order (from the early year to the more recent year).

* Three new references must be included:

1. Rodriguez-Merchan EC. Surgical treatment of isolated patellofemoral osteoarthritis. HSS J 2014;10:79-82.

2. Rodriguez-Merchan EC. Evidence based conservative management of patello-femoral syndrome. Arch Bone Jt Surg 2014;2:4-6.

3. Rodriguez-Merchan EC. The present situation of patellofemoral arthropalsty in the management of solitary patellofemoral osteoarthritis. Arch Bone Jt Surg 2020;8:325-331.

* In lines 40 and 57 add the aforementioned Reference 2.

* In line 58 add add the aforementioned references 1 and 3.

* In line 59 change "physiotherapy" for "physical and rehabilitation medicine".

* In line 279 add the aforementioned references 1, 2 and 3.

* In line 314 add the aforementioned reference 3.

* In line 314 add the aforementioned reference 2.

Author Response

The systematic review is very interesting and well done. However I recommend the following changes to the authors to improve the quality of their paper:

Thank you for your time and helpful feedback.

Point 2.1 * CHANGE THE TITLE: My suggestions is "Non-surgical and surgical treatments for patellofemoral pain and patellofemoral osteoarthritis: A systematic review.

Response 2.1: Thank you for the thoughtful suggestion. We understand that this suggestion likely refers to Point 2.2 below regarding the term ‘nutraceutical’ and we are happy to reconsider the title of our paper to reflect this comment. Our concern regarding the suggested new title is that ‘non-surgical’ would also include treatments such as therapeutic exercise, massage therapy, ultrasound, and more. Therefore we would like to suggest the following new title that we believe still captures the intention of our review question, and hope that Reviewer 2 finds that this addresses the spirit of the suggestion.

Changes made 2.1: New title – “Medical interventions for patellofemoral pain and osteoarthritis: a systematic review”

Point 2.2 * Delete the term "nutraceutical" all along the paper, tables and figures (it is misleading).

Response 2.2: Thank you for this suggestion. We understand Reviewer 2’s point and agree that the current use of the term is potentially misleading. However, our search was designed to capture any RCT investigating pharmaceuticals, nutraceuticals or surgical interventions for PFP and PF osteoarthritis. While very few nutraceuticals were found through our search (e.g., olive oil, glucosamine, hyaluronic acid), we have included them in our study and would therefore prefer to label them appropriately. Moreover, because some of them can only be administered in a medical setting (i.e. through injection), we have combined the pharmaceutical and nutraceutical into a single section rather than presenting them separately. To address Reviewer 2’s concerns, we have made the following changes to the manuscript to clarify the use of this term.

Changes made 2.2: Page 2, line 59 – “In addition to pharmaceutical agents, nutraceuticals are reported to be of therapeutic benefit in certain musculoskeletal conditions [28]. They are derived from dietary sources, can be administered in different formats (e.g. taken orally or by injection), and are regulated differently in different countries. Therefore, nutraceuticals can be prescribed and administered in clinical settings, but in many countries patients can also obtain nutraceuticals off-the-shelf and taken without specific recommendations by a medical practitioner. Finally, in cases where patients have not responded favorably to non-surgical care, clinicians may also refer them for surgical consults (10 to 12%) [12,22,29,30].”

Point 2.3 * In Figure 1, Table 2, Table 3 and Table 4, put the references in chronological order (from the early year to the more recent year).

Changes made 2.3: Figure 2, Table 2 and Table 3 have been formatted in chronological order as suggested, though we have done so within each of our two broad categories of intervention for ease of interpretation. Table 4 is now also formatted chronologically as requested (highlighted yellow because changes not tracked in word table).  

Point 2.4 * Three new references must be included:

  1. Rodriguez-Merchan EC. Surgical treatment of isolated patellofemoral osteoarthritis. HSS J 2014;10:79-82.
  2. Rodriguez-Merchan EC. Evidence based conservative management of patello-femoral syndrome. Arch Bone Jt Surg 2014;2:4-6.
  3. Rodriguez-Merchan EC. The present situation of patellofemoral arthropalsty in the management of solitary patellofemoral osteoarthritis. Arch Bone Jt Surg 2020;8:325-331.

* In lines 40 and 57 add the aforementioned Reference 2.

* In line 58 add add the aforementioned references 1 and 3.

* In line 279 add the aforementioned references 1, 2 and 3.

* In line 314 add the aforementioned reference 3.

* In line 314 add the aforementioned reference 2.

Response 2.4: Thank you for these suggested reviews and overviews. As requested, we have added these 3 references to the manuscript in areas relating to their relevance regarding interventions.

Point 2.5 * In line 59 change "physiotherapy" for "physical and rehabilitation medicine".

Response 2.5: We have changed this line as suggested, it is now line 68.

Reviewer 4 Report

Thank you for your effort. You performed extensive search about vague territory. However, there is some lacking to be generally consensused.

I give some specific comments.

Key words: chang order as " Patellofemoral pain, Patellofemoral osteoarthritis, Pharmaceuticals, Nutraceuticals, Surgery.

First important drawback is definition. There are many causes for patellofemoral pain, but you addressed all together. Your methodology is great, but it is from wrong definition.

Second, most results are very limited evidence.

I recommend to specifiy the disease category as patellofemoral pain syndrome. PFPS and PFP are different.

You reached some evidence in patellofemoral arthroplasty and TKRA. However, I don't think that is is right.

Author Response

Reviewer 3

Thank you for your effort. You performed extensive search about vague territory. However, there is some lacking to be generally consensused.

I give some specific comments.

Thank you for your time and helpful feedback.

Point 3.1: Key words: chang order as " Patellofemoral pain, Patellofemoral osteoarthritis, Pharmaceuticals, Nutraceuticals, Surgery.

Response 3.1: We have changed the order of keywords as suggested.

Point 3.2: First important drawback is definition. There are many causes for patellofemoral pain, but you addressed all together. Your methodology is great, but it is from wrong definition.

Response 3.2: Thank you for this suggestion. We agree with Reviewer 3 that there are many causes for patellofemoral pain, which definitely makes this field very complex but interesting. We also acknowledge that terminology has changed over the past decades, and also continues to differ among medical disciplines. Moreover, emerging research is starting to suggest that many conditions causing PFP may lie on a pathological spectrum varying from PFP in the absence of structural disease, through to chondromalacia patella and, in later ages, patellofemoral osteoarthritis. While this spectrum is still being investigated, we wished to be intentionally inclusive in this review to ensure that these possibly-related conditions are not being investigated in silos. To navigate this, we have chosen to use terminology currently recommended by the International Patellofemoral Research Network (Crossley et al. 2016, reference #1) and have included patellofemoral OA in our study. We have acknowledged this in the limitations section of our discussion, however to further acknowledge this, we have made the following changes:

Changes made 3.2:  Page 2, line 90: “We included studies investigating PFP using intentionally broad criteria based on current recommended terminology [1]. Specifically, for PFP we included individuals diagnosed either symptomatically or structurally, and as such included synonyms for PFP such as patellofemoral pain syndrome and chondromalacia patella, recognizing that terminology differs among different disciplines and has also changed over time [1,36]. We also included individuals with PFOA because emerging evidence suggests that many conditions affecting the patellofemoral joint may lie along a pathological spectrum, with PFOA potentially representing a late stage of chronic PFP (even though treatment approaches may currently differ) [7,8,16-18].”

Point 3.3: Second, most results are very limited evidence.

Response 3.3: Thank you for recognizing the importance of this review. Indeed, our findings suggest that clinical decisions regarding interventions are currently being made with very little supporting evidence. This is an important message and we hope this review will stimulate further, more rigorous research in this area to address this important knowledge gap.

Point 3.4: I recommend to specify the disease category as patellofemoral pain syndrome. PFPS and PFP are different.

Response 3.4: Thank you for this suggestion. As described above, we have decided to use terminology currently recommended by the International Patellofemoral Research Network (Crossley et al. 2016, reference #1). This group defines PFP as the umbrella term, and defines it as a synonym for other terms including: PFP syndrome; chondromalacia patella; anterior knee pain and/or syndrome; and runner’s knee. To address this comment and clarify our terminology, we have added clarifying details to the manuscript’s Method section.

Changes made 3.4: Please see changes made 3.2.

Point 3.5: You reached some evidence in patellofemoral arthroplasty and TKRA. However, I don't think that is right.

Response 3.5: Thank you for this comment. We suspect this comment may relate to our definition of long-term outcomes as anything greater than 6 months. We can see how this could be problematic for surgical outcomes, and have acknowledged this in our Discussion section. However, to further address this comment, we have added some detail to both the Results and Discussion section to be more clear about the period of follow-up of both studies. This is very relevant for this type of intervention and we thank Reviewer 3 for bringing this to our attention.

Changes made 3.5: Page 16, line 293 – “In one study [40], short and medium-term results favored patellofemoral arthroplasty for two pain subscales (KOOS Symptoms, SMD 0.6 [0.2, 1.0]; and SF-36 Bodily Pain, SMD 0.5 [0.1, 0.9]) but results did not differ for remaining KOOS and SF-36 subscales. In the same study, long-term results (2 years post-surgical) only remained in favor of patellofemoral arthroplasty for KOOS Symptoms (SMD 0.5 [0.1, 0.9]). The other study reported no differences in any outcome up to 1 year after surgery, including several outcomes measured five years after surgery [43].”

Page 17, line 328 – “This evidence suggests possible early benefits in some symptoms compared to total knee arthroplasty for treating isolated PFOA, but likely no differences between interventions one to five years following surgery, suggesting that both approaches may be equally effective during this period of follow-up.”